# Effectiveness of A Nurse-Led Multimodal Intervention in Preventing Blood Culture Contamination: A Before-and-After Study

**DOI:** 10.3390/healthcare12171735

**Published:** 2024-08-31

**Authors:** Susana Filipe, Teresa Martins, Paulo Santos-Costa, Filipe Paiva-Santos, Amélia Castilho, Celeste Bastos

**Affiliations:** 1Health Sciences Research Unit: Nursing (UICISA: E), Nursing School of Coimbra (ESEnfC), 3000-232 Coimbra, Portugal; afilomena@esenfc.pt; 2Local Health Unit of Baixo Mondego, 3094-001 Figueira da Foz, Portugal; 3Centre for Health Technology and Services Research, Health Research Network (CINTESIS@RISE), Nursing School of Porto (ESEP), 4200-072 Porto, Portugal; teresam@esenf.pt (T.M.); cbastos@esenf.pt (C.B.); 4Nursing Research, Innovation and Development Centre of Lisbon (CIDNUR), Nursing School of Lisbon (ESEL), 1600-096 Lisbon, Portugal; p.costa@esel.pt; 5Instituto de Saúde Ambiental (ISAMB), Faculdade de Medicina, Universidade de Lisboa, 1649-026 Lisbon, Portugal

**Keywords:** blood culture, contamination, multimodal intervention, risk assessment, nursing, antimicrobial stewardship, patient safety, quality improvement

## Abstract

Blood culture is crucial for accurate and timely bacteremia diagnosis and guide antibiotic therapy. However, during culture sampling, contamination can occur, resulting in misdiagnosis, unnecessary antibiotic exposure, and prolonged hospitalization. This before-and-after intervention study aimed to evaluate the effectiveness of a multimodal intervention in preventing blood culture contamination. The study was conducted in a 170-bed hospital in Portugal and included a total of 23,566 blood cultures. Contamination rates were assessed in two phases: Phase 1 (before intervention, month 0) included 10,928 cultures, and Phase 2 (after intervention, month 6) included 12,638 cultures. During the study period, a multimodal intervention targeting the nursing staff was implemented, consisting of training actions, guideline updates, regular data monitoring and feedback, and introduction of a blood culture pack. Following the intervention, blood culture contamination decreased from 6.8% (Phase 1) to 3.9% (Phase 2). A comparative analysis revealed that the risk of contamination before the intervention was nearly four times higher in first culture, OR = 3.97 (CI 2.86–5.49). Our findings suggest that the multimodal intervention enhanced nurses’ adherence to recommended practices, resulting in a reduced risk of blood culture contamination, earlier identification of infectious agents, and improved accuracy of antibiotic therapy.

## 1. Introduction

Antimicrobial resistance is an emerging public health threat that will worsen if no action is taken [1]. To mitigate the further dissemination of this “silent pandemic”, it is imperative to implement measures that minimize the unnecessary use of antibiotics, one of the key drivers of antimicrobial resistance. A blood culture is an invaluable diagnostic tool, providing crucial information that enables the identification of bacteremia, thereby guiding the administration of appropriate antibiotic therapy. However, the reliability of culture results depends on compliance with blood culture sampling procedures, as non-compliance may result in inconclusive or erroneous outcomes due to contamination [2]. As blood is sterile, positive blood cultures with a known pathogen have a high positive predictive value for infection [3], indicating the infectious cause of the disease and informing antimicrobial therapy. However, false-positive blood cultures are frequently encountered due to specimen contamination [4], leading to misinterpretation and negatively impacting both patients and the healthcare system.

One of the most significant consequences of contaminated blood cultures is misdiagnosis, which can result in the prescription of inappropriate antibiotics. This, in turn, increases the patient’s exposure to antibiotics, thereby elevating the risk of toxicity and antimicrobial resistance [3,4,5]. This is particularly the case when coagulase-negative staphylococci contaminants are only susceptible to antibiotics with higher toxicity potential, such as vancomycin [6]. Bates et al. [7] observed that the incidence of intravenous antibiotic courses was 39% higher in cases of contaminant blood cultures than in those of culture-negative patients. Similarly, Lee et al. [8] found that 41% of patients with contaminants received unnecessary intravenous antibiotics. Augmented antibiotic exposure is known to expose patients to adverse effects including allergies and the disruption of the host microbiome, leading to *Clostridioides difficile* infection [7].

The American Society for Microbiology (ASM) and the Clinical and Laboratory Standards Institute (CLSI) set a benchmark for each healthcare institution to maintain a blood culture contamination (BCC) rate below 3% [4]. However, reported contamination rates vary considerably, ranging from 0.6% to 12% [9,10,11]. Economic health costs associated with antimicrobial therapy, additional laboratory and microbiology costs, and increased length of stay must also be considered [12]. Therefore, BCC has an impact on the efficiency, safety, and quality of healthcare provision [3,13]. The impact of nursing practices on antibiotic decisions may vary depending on the local context and prevailing cultural norms. One potential avenue for this influence is through the implementation of accurate microbiological testing practices [14]. Blood culture sampling is a procedure commonly performed by nurses worldwide, integral to their role within the healthcare team. Being responsible for the timely and appropriate sampling of microbiological specimens such as blood cultures, nurses are presented with an additional opportunity to support antimicrobial stewardship objectives [15].

While BCC can occur in the laboratory, such events are rare [16], and contamination usually happens during blood sampling by healthcare professionals. During venipuncture, bacteria on skin can dislodge into the specimen [3]. Several factors may contribute to this risk, including healthcare professionals’ insufficient knowledge or technical skill, blood draws from intravascular lines, and a lack of organizational feedback on BCC rates by each specific setting/department [17]. Additionally, a stressful working environment, such as in emergency departments, and patient-related factors, such as end-stage renal disease and older age, have been linked to increased rates of BCC [18]. This is attributed to the fact that these patients are potential carriers of resistant strains of skin commensals due to their repeated visits to healthcare facilities and the presence of poorly accessible veins [18].

The implementation of preventive measures has the potential to mitigate this risk. Previous studies have demonstrated the efficacy of educational interventions targeting nursing staff in intensive care units [11,19,20], emergency departments [17], and hospital-wide [21], while others included other healthcare professionals [16,22,23]. The introduction of standardized blood culture packs have also yielded positive results in a hospital-wide intervention [22] and in emergency departments [23]. These findings suggest that interventions targeting multiple elements through an integrated and structured approach, such as multimodal strategies, may lead to more effective behavior changes and improve care outcomes related to infection prevention and control [11,24].

To the best of our knowledge, previous studies have not focused on the long-term impact of multimodal interventions for preventing BCC in acute care settings [21,25]. Thus, our study aimed to evaluate the effectiveness of a nurse-led multimodal intervention in preventing BCC in a secondary hospital.

## 2. Materials and Methods

### 2.1. Design

A before-and-after intervention study was conducted in a 170-bed secondary hospital, encompassing 23,566 blood cultures, processed by the microbiology laboratory between 1 January 2019 and 16 June 2024. Each set consisted of two culture bottles, aerobic and anaerobic. For children, one culture bottle was used. At this healthcare institution, blood cultures are carried out by nurses rather than by phlebotomy teams. The BCC rates were evaluated at two different phases: phase 1, before the intervention, included 10,928 cultures collected over a 29-month and 16-days period; phase 2, after the intervention, included 12,638 cultures collected over a 29-month and 16-days period. The multimodal intervention spanned a six-month period (from 16 June 2021 to 31 December 2021) with an additional intervention in December 2022. This study was approved by the institutional Ethics Committee. The intervention is reported following the Template for Intervention Description and Replication (TIDieR) checklist [5].

### 2.2. Sample

The blood sampling technique, microbiological testing methods for diagnosing bloodstream infections or BCC, and the reporting procedures were standardized according to the microbiology laboratory’s protocols. The standard sampling technique for blood cultures involved obtaining a minimum of two consecutive blood culture sets, with samples collected from different sites (i.e., by phlebotomy or central venous catheter). In the latter, the recommendation was to draw a blood culture set by phlebotomy as well. Each blood culture set consisted of one aerobic BacT/ALERT FA (bioMérieux, Marcy l’Étoille, France) culture and one anaerobic BacT/ALERT FN (bioMérieux) culture, containing up to 10 mL of blood each. For children, a BacT/ALERT PF (bioMérieux) culture was used, containing up to 4ml of blood. After collection, the blood culture bottles were kept at room temperature and sent to the microbiology laboratory within one hour. Blood cultures were incubated in BacT/ALERT^®^ 3D (bioMérieux) for 5 days (mean) or until microbial growth was detected. In exceptional cases, such as those involving endocarditis or fastidious microorganisms, the incubation period was found to be prolonged. The microbiology laboratory information system (MaxData 5.3.5) was used to automatically register the detection of microbial growth, which was then followed by Gram staining, a 24-h culture, and finally, the identification of microorganisms and an antibiotic sensitivity test.

A blood culture was deemed contaminated if one or more of the following organisms were identified in only one of a series of blood culture specimens collected from a single patient on the same day: coagulase-negative staphylococci, *Micrococcus* species, *Propionibacterium* species, *Corynebacterium* species and *Bacillus* species [26]. Despite recommendations to the contrary, single blood culture sets were sent for microbiological analysis. Any single positive blood sample with isolated natural microbiota was flagged for discussion with the patients’ physician. If no clinical signs of infection were present, the culture was considered as contaminated. To calculate the BCC rate, we used the following Formula [27]:(1)$$BCC\;rate=\frac{number\;of\;contaminated\;cultures}{total\;number\;of\;cultures}\times 100$$

All blood cultures performed at the institution, regardless of the time of the day, were included in the study.

### 2.3. Intervention Overview

The challenge of the BCC was addressed by the infection prevention and control (IPC) lead nurse during a meeting with the hospital’s head nurses and IPC link nurses on 16 June 2021. The aim was to raise awareness about BCC rates and their implications for patient safety. It was noted that there was often non-compliance with the established protocols for skin antisepsis and phlebotomy procedures, including the inadequate use of available resources. Specifically, there were no recommendations in place for preparing necessary material on a sterile drape or disinfecting the tops of culture bottles. During this meeting, several factors potentially contributing to the current BCC rates were discussed, including challenging work environments, outdated information sources, and the need for more comprehensive information and feedback from management and IPC leaders.

As a direct result of this meeting, the IPC team and the Microbiology department undertook a comprehensive review of the institutional protocols for blood culture and associated resources. Additionally, a complete training program for nurses was developed, as depicted in Figure 1.

Following a comprehensive review of the existing literature on blood culture sampling practices, institutional guidelines were revised to align more closely with the current evidence [2,4]. The recommendations were structured into three distinct steps of the blood culture collection procedure:Before blood draw: (1) confirm patient’s identification, explain the rationale for the procedure, and obtain consent; (2) perform hand hygiene with hand rub solution (73.4 g ethanol and 10.0 g propan-2-ol); (3) assemble necessary material to avoid further interruptions during the procedure; (4) verify the bottles’ expiry date and bottoms’ color to ensure the culture quality; (5) clean the work surface; (6) perform hand hygiene with hand rub solution (73.4 g ethanol and 10.0 g propan-2-ol); (7) prepare the material on a sterile drape; (8) remove the bottle caps, disinfect the top with 2% Chlorhexidine in 70% alcohol, and allow it to dry;During blood draw: (1) apply a single use tourniquet; (2) perform skin antisepsis with 2% Chlorhexidine in 70% alcohol and allow it to dry; (3) perform hand hygiene with hand rub solution (73.4 g ethanol and 10.0 g propan-2-ol), and put on sterile gloves; (4) perform phlebotomy with a Safety S-Monovette^®^ needle and connect the membrane adapter to a 20 mL syringe (closed blood culture system); (5) pinch each culture bottle with one S-Monovette^®^ safety needle and adapt the syringe, inoculating 10mL of blood in each culture bottle;After blood draw: (1) clearly identify the culture bottles; (2) send the culture bottles to the laboratory within 1 h; (3) document the procedure in patient’s health record.

The procedure was disseminated in a written format, as a poster (available as Appendix A), and in a three-minute video (accessible in its original format via https://www.powtoon.com/c/c1Hkia1Ao8r/1/m, accessed on 11 June 2024). These resources were prepared in collaboration with the IPC link nurses who volunteered.

Furthermore, a 30-min structured presentation was developed for use by IPC link nurses within their respective teams. The primary objective of this presentation was to enhance nurses’ awareness of BCC impact and improve their knowledge and skills. The presentation covered several key messages including the definition of blood culture, its purpose, and significance. It addressed the implications of BCC and provided guidance on the optimal timing for blood culture collection, recommending three sets of blood culture samples, each with a volume of 20 mL (10 mL in each culture). The presentation also specified appropriate collection sites and detailed the procedure for skin antisepsis and blood culture collection. Additionally, it emphasized the importance of seeking assistance to mitigate the risk of BCC during the blood draw procedures, especially for patients with difficult venous access or those unable to cooperate.

From September 2021 to December 2021, the IPC link nurses delivered 20 presentations across nine wards, including the emergency ward, intermediate care unit, orthopedic ward, internal medicine ward, medical specialties ward, general surgery ward, surgical specialties ward, and pediatrics. This healthcare institution does not have an intensive care unit. Approximately 85% of all nurses attended these presentations.

In 2022, the IPC lead nurse confirmed that the posters were available in all wards and conducted informal discussions with the nursing teams and IPC link nurses to address any remaining questions or concerns about the blood culture sampling procedure. IPC link nurses provided just-in-time coaching to their peers, and feedback on the BCC rates was shared with the team on a quarterly basis. Additionally, In December 2022, an additional intervention was introduced, proposed by the IPC link nurses from the medical specialties ward. To streamline the preparation of all required material for the blood culture sampling procedure, a standardized pack was developed and implemented across all wards in collaboration with the IPC link nurses. The packs are assembled in the wards following a protocol devised by the research team and include the following materials:1 blood culture set (1 aerobic bottle, BacT/ALERT FA, and 1 anaerobic bottle, BacT/ALERT FN; for children, 1 bottle BacT/ALERT PF);2 sterile gauze 5 cm × 5 cm packages;1 sterile drape;1 clean drape;3 S-Monovette^®^ safety needles (SARSTEDT, Nümbrecht, Germany);1 Safety-Multifly^®^ needle (SARSTEDT);1 Multi-adapter, Luer (SARSTEDT);1 Membrane adapter (SARSTEDT);1 sterile 20 mL syringe;1 single-use tourniquet;1 sterile dressing;1 leaflet with the blood culture steps.

The implemented multimodal intervention [5], depicted in Figure 2, included the following elements:System change (build it): improved use of available resources and the introduction of a blood culture pack;Training and education (teach it): the issue of BCC rates was discussed with the head nurses and IPC link nurses. In collaboration with the Microbiology Department, a comprehensive training program for nurses was devised and implemented;Monitoring and feedback (check it): the BCC rate was monitored, and feedback was provided on a quarterly basis;Reminders and communication (sell it): institutional guidelines were revised and disseminated in written format, as a poster format, and in a three-minute video;Culture change (live it): The IPC link nurse’s role was to disseminate the procedure for blood culture sampling and ensure its sustainability. The IPC lead nurse was present throughout the duration of the project, continuously accompanying the ward nurses.

### 2.4. Statistical Analysis

The data was analyzed using IBM-SPSS, version 29, and a significance level of 0.05 was used. Descriptive statistics were used to summarize the data through rate percentage, mean (M), and standardization deviation (SD). In the analysis of differences in means between two independent continuous variables, we used the Student’s *t*-test. Odds Ratio (OR) was calculated to quantify the strength of the association between the intervention and the reduction in BCC rates, providing a measure of the effect size. Results were reported with corresponding 95% confidence intervals (CI). The OR analyses were classified into cases and controls. Regarding the study of the impact of the multimodal intervention, the reference periods were from January 2019 to 16 June 2021 (cases) and from January 2022 to 16 June 2024 (controls), with the period during intervention being excluded. Regarding the study of the impact of the introduction of the blood culture pack, the periods were from January 2019 to 16 June 2021 and from January 2022 to December 2022 (cases), and from January 2023 to 16 June 2024 (controls). Graphical representations, such as line graphs, were employed to illustrate the trends and differences in BCC rates across the study period.

## 3. Results

Over the course of the entire study period, 23,566 blood cultures were obtained, of which 1230 were found to be contaminated. This equates to a BCC rate of 5.2% for the entire study period, excluding the six-month multimodal intervention. Phase 1 of the study corresponds to the period before the intervention. Phase 2, which occurs after the intervention, is divided into two periods: Moment 1, which takes place after the training actions for IPC link nurses and nursing staff, and Moment 2, which takes place after the introduction of the standardized packs.

Of the 23,566 blood cultures, 10,928 were taken in Phase 1 (January 2019 to 16 June 2021, a 29 months and 16 days period) and 12,638 cultures were taken in Phase 2 (January 2022 to 16 June 2024, a 29 months and 16 days period). Contamination rates during Phase 1 were 6.8%, which significantly decreased to 3.9% by the end of Phase 2 (Table 1).

The BCC reduction along the study is presented in the line graph (Figure 3).

The mean contamination values for samples across all sets decreased significantly after the six-month intervention. Statistically significant reductions were observed in both Moment 1 (training actions) and Moment 2 (introduction of the standardized pack). The p-values for these reductions were consistently below 0.05, indicating strong statistical significance, particularly in Set 1 and Set 2 cultures (Table 2).

A comparative analysis was conducted on the contamination of three sets of blood cultures. Each blood culture set consisted of one aerobic BacT/ALERT FA (bioMérieux) culture and one anaerobic BacT/ALERT FN (bioMérieux) culture, containing up to 10 mL of blood each. The analysis aimed to assess the risk of contamination over two distinct periods: Phase 1, from 1 January 2019 to 16 June 2021, and Phase 2, from 1 January 2022 to 16 June 2024. Phase 1 served as a baseline for comparison, representing the pre-intervention moment, while Phase 2 was used to assess the impact of a multimodal intervention with an additional intervention during the study. The results are presented in Table 3.

To ascertain the extent of contamination risk associated with the samples prior to and following the implementation of the pack (which contains the necessary materials for blood cultures collection), an OR analysis was conducted, comparing all the analyses conducted between 1 January 2019 and 16 June 2021, with the period from 1 January 2023 to 16 June 2024 (Table 3).

To compare the risk of contamination after the intervention, Phase 2, an OR analysis was conducted to compare two time periods: the initial period (Moment 1), which spanned from 1 January 2022 to 31 December 2022, and the subsequent period (Moment 2), which started with the introduction of the multimodal intervention and continued to 16 June 2024 (Table 3).

A comparison of the results from culture 1 before and after the multimodal intervention reveals a significant reduction in the risk of obtaining contaminated blood cultures. Prior to the intervention, the OR was 2.23 (95% CI 1.77–2.80), indicating a twofold increased likelihood of contamination compared to the post-intervention period. Furthermore, the remaining cultures also exhibited a higher risk for contamination before the intervention, although slightly lower.

A comparison of risk contamination before the multimodal intervention and after the pack introduction reveals an OR value of 3.97 (95% CI 2.86–5.49) for the collection of culture 1. This indicates that the risk of contamination was nearly four times higher before the implementation of the change in structure and process.

A comparison of the two moments from Phase 2, Moment 1 (prior to pack introduction) and Moment 2 (after pack introduction), reveals an OR value of 3.49 (95% CI 2.406–5.07) for the collection of culture 1. This corresponds to a risk of contamination that is more than three times higher before pack introduction compared to the post-intervention period without this structural change. For the remaining cultures, the OR value decreased, although the risk remained higher before than after the implementation of this measure.

## 4. Discussion

Our results suggest that the implementation of a multimodal intervention, which included components addressing structural (e.g., available resources) and process-related (e.g., nurses’ care delivery practices) [28] challenges, can significantly reduce BCC rates. In this study, a nurse-led intervention that included a comprehensive training program, updates to information sources and institutional protocols, regular personalized staff feedback, and the introduction of a blood culture pack, promoted sustainable changes. These efforts led to a sustained reduction in BCC rates, bringing them to 2.3% in the first trimester of 2024, below the benchmark standard of 3% set by the ASM and CLSI [4].

Education-based interventions have shown to be effective in preventing BCC. Kumthekar et al. [11] found that providing evidence-informed training to nursing staff resulted in a significant reduction of the BCC rates from 6.16% to 3.30%. Similarly, Sánchez-Sánchez et al. [20] reported that an education-based intervention not only reduced BCC rates from 8% to 3.5%, but also resulted in significant savings for the healthcare system. This financial impact is substantial, as the estimated direct and indirect costs associated with each contaminated culture can exceed £5000 [12,29].

While education-based interventions are crucial in reducing BCC rates, it is important to note that the introduction of the blood culture pack was a key component of this intervention. A comparable approach was used in the study conducted by Dhillon et al. [22], which found that implementing a blood culture pack alongside a formal training program and increased institutional awareness of BCC resulted in a significant reduction in contamination rates from 8.7% to 3.0%. In our study, similar efforts were also focused on enhancing awareness and actively involving frontline nurses in driving the change, which was identified as a critical factor in the success of the multimodal intervention.

The initial meeting with the head nurses and IPC link nurses established an environment supportive of practice transformation. By presenting both global and departmental BCC rates, the meeting fostered a unified approach among the nursing staff, aiming to achieve BCC rates that meet the ASM and CLSI benchmark standards. Following the education-based intervention, a system of reminders regarding blood culture technique was implemented, as well as the regular communication of BCC rates. A similar methodology was utilized in Su’s study [30], which led to a reduction in healthcare-associated infections. The authors encouraged nurses to contribute their ideas and proposals for improvement, which, in our view, facilitated adherence to recommended changes and enhanced staff involvement over time. According to Gesser-Edelsburg et al.’s study [27], encouraging the participation of professionals is an effective method for promoting change and reducing healthcare associated infections, as it allows for the exploration of creative solutions to problems that may not be addressed in guidelines.

In our study, the close monitoring of various wards by the IPC lead nurse, along with regular positive feedback on progress, aligns with the principles of the Hawthorne effect. This effect suggests that behavior change can occur as a motivational response to attention received through external observation and evaluation [31,32]. This concept supports Eckmanns et al.’s [32] recommendation for more frequent presence of IPC teams in hospital wards, which encourages active nurse involvement. In our study, this approach led to the introduction of the blood culture pack during Phase 2.

In addition to training and regular feedback, our findings demonstrate a fourfold reduction in the risk of contamination following the introduction of the blood culture pack, compared to the pre-intervention period. The introduction of the blood culture pack across various wards appears to have supported nurses in aligning their care delivery with current evidence-based standards, further reducing the already lower BCC rates observed during the initial months of Phase 2 (e.g., training actions and updates to information sources). These findings suggest that improving the quality and accessibility of existing materials enhances nursing practices and helps sustain behavior changes over the long term. In our study, the pack was developed using materials already available at the hospital, with standardized packaging and labeling minimizing variations in frontline nurses’ practices. Following the introduction of the pack, BCC rates approached the benchmark of 3% [4]. This strategy can be viewed as a nudge intervention [33].

Another significant and widely discussed consequence of BCC is the unnecessary use of antibiotics, which increases the risk of antimicrobial resistance [12,29]. Nurses must be aware of this broader context when performing their professional duties. Accurate blood culture results are crucial for determining the most appropriate antimicrobial therapy in cases of suspected or confirmed bloodstream infections. The quality of microbiology culture directly impacts the accuracy of microbiology reports, making BBC a key quality indicator, according to the Centers for Disease Control and Prevention (CDC) [3].

The CDC’s Core Elements of Hospital Antibiotic Stewardship Programs highlights the optimization of microbiology cultures as nursing-based interventions in antimicrobial stewardship programs [14,34]. Nurses are involved in health promotion, education, and coordination of patient care, collaborating with a range of other healthcare professionals in antimicrobial stewardship activities, albeit often in an unrecognized and unintegrated manner [28,35].

Although nurses may not always be aware of their role in antimicrobial stewardship, it is reasonable to consider the BCC rate as a quality outcome related to nursing interventions. This aligns with Irvine et al.’s [28] Nursing Role Effectiveness Model, which explores the relationships between healthcare structural variables, nurses’ roles, and their impact on patient and system outcomes. Linking nursing interventions to patient outcomes and evidence-based practices has the potential to transform nursing practice and contribute to significant antimicrobial stewardship efforts [36].

### Limitations

Multimodal interventions are recognized by the World Health Organization as a core component for effective IPC programs, essential for fostering a culture that ensures the long-term sustainment of practice changes [37]. Van Buijtene and Foster [38] note that improvements in IPC performance attributed to the Hawthorne effect may diverge from actual behavior over extended periods. However, our study demonstrated that nurses’ clinical practice continued to improve over the 29 months following the beginning of the multimodal intervention. While our finding underscores the importance of consistent follow-up to ensure the long-term sustainability of change over time [38], several study limitations must be considered.

The absence of patient demographic data may introduce bias into the study, and the variability among nursing teams across different wards could influence the results. As a consequence, our calculations focus on culture bottles rather than sets, which could represent different values for BCC rates. During the intervention phase, the research team did not assess the specific skills and knowledge acquired by nurses through the training actions. However, the observed decrease in the BCC rate may serve as an indirect indicator of enhanced nurse competency in blood culture sampling. Cost analysis was not performed in this study. However, it is reasonable to conclude that the reduction in BCC likely resulted in cost savings for the institution. These savings would stem from avoiding additional examinations, reducing the need for antibiotic prescriptions, and shortening the length of hospital stays [12,29]. Finally, the study design does not allow for the establishment of causality or the control of all potential confounding variables that might affect the results found.

## 5. Conclusions

Study findings suggest that multimodal interventions have a long-term impact on reducing BCC. Interventions that address both structural and procedural challenges, including training needs and quality care resources, are more effective at significantly lowering BCC rates. The direct and sustained involvement of both head and frontline nurses was crucial for the successful long-term implementation of the multimodal intervention. This underscores the importance of nurse engagement and regular monitoring by infection prevention and control leadership in enhancing the quality of nursing care.

## Figures and Tables

**Figure 1 healthcare-12-01735-f001:**
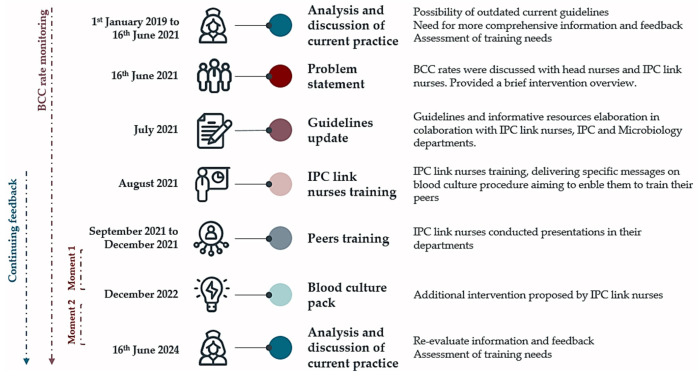
Timeline of the developed multimodal intervention.

**Figure 2 healthcare-12-01735-f002:**
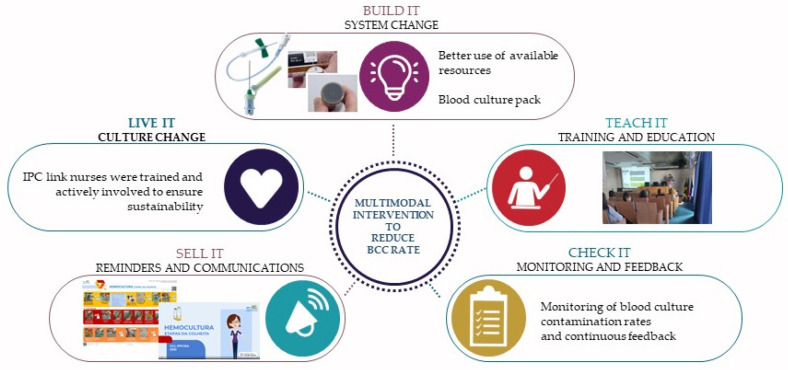
Overview of the implemented multimodal intervention.

**Figure 3 healthcare-12-01735-f003:**
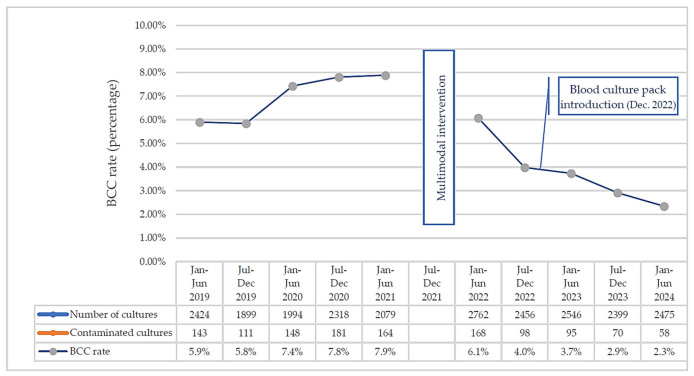
BCC rate evolution through the study period, presented by trimester in study’s Phase 1 (January 2019–16 June 2021) and Phase 2 (January 2022–16 June 2024).

**Table 1 healthcare-12-01735-t001:** BCC rates in Phase 1 (before the intervention) and Phase 2 (after the intervention, detailing rates at different moments).

	Phase 1	Phase 2		
	1 January 2019–16 June 2021	Moment 1 ^1^	Moment 2 ^2^	Moment 1 and Moment 2
		1 January 2022–31 June 2022	1 January 2023–16 June 2024	
Total number of cultures	10,928	5218	7420	12,638
Contaminated cultures	741	266	223	489
BCC rate	6.80%	5.10%	3.00%	3.90%

^1^ Moment 1—the period of time in phase 2, after the six-month multimodal intervention and before the pack introduction. ^2^ Moment 2—the period of time in phase 2, after the pack introduction.

**Table 2 healthcare-12-01735-t002:** Impact of Multimodal Intervention on Blood Culture Contamination Rates: Before-and-After Analysis.

	Multimodal Intervention ^1^ Phase 1 and Phase 2			After Multimodal Intervention ^2^ Moment 1 and Moment 2		
	Before	After	*p* Value	Before	After	*p* Value ^3^
Set 1 culture 1 M(SD) culture 2 M(SD)	0.07 (0.25)	0.03 (0.18)	<0.001	0.06 (0.23)	0.02 (0.013)	<0.001
	0.08 (0.27)	0.04 (0.20)	<0.001	0.06 (0.24)	0.03 (0.17)	<0.001
Set 2 culture 1 M(SD) culture 2 M(SD)	0.07 (0.25)	0.05 (0.21)	0.004	0.06 (0.24)	0.03 (0.18)	0.001
	0.07 (0.26)	0.05 (0.21)	<0.001	0.06 (0.23)	0.04 (0.19)	0.018
Set 3 culture 1 M(SD) culture 2 M(SD)	0.07 (0.26)	0.04 (0.20)	0.008	0.05 (0.21)	0.04 (0.19)	0.597
	0.07 (0.25)	0.05 (0.21)	0.028	0.05 (0.22)	0.05 (0.21)	0.779

^1^ Multimodal intervention: Phase 1, before intervention (1 January 2019 to 16 June 2021) and Phase 2, after intervention (1 January 2022 to 16 June 2024). ^2^ After multimodal intervention (Phase 2): Moment 1, before blood culture pack (January 2022 to December 2022) and Moment 2, after blood culture pack (January 2023 to 16 June 2024). ^3^ Student’s *t*-test.

**Table 3 healthcare-12-01735-t003:** Odds ratio and confidence intervals for BCC before and after intervention, across the different study phases.

		Multimodal Intervention ^1^ Phase 1 (1 January 2019/16 June 2021) and Phase 2 (1 January 2022/16 June 2024)		Blood Culture Pack ^2^ Phase 1 (1 January 2019/15 June 2021) and M 2 (1 January 2023/16 June 2024)		After Multimodal Intervention ^3^ M 1 (1 January 2022/31 December 2022) and M 2 (1 January 2023/15 June 2024)	
		OR	IC95%	OR	IC95%	OR	IC95%
Set 1	Culture1	2.23	1.77–2.80	3.97	2.86–5.49	3.49	2.40–5.07
	Culture 2	1.98	1.55–2.54	2.69	1.92–3.75	2.17	1.46–3.23
Set 2	Culture 3	1.46	1.13–1.89	1.97	1.43–2.72	1.90	1.30–2.79
	Culture 4	1.69	1.31–2.17	1.94	1.42–2.66	1.61	1.09–2.36
Set 3	Culture 5	1.72	1.14–2.58	1.67	1.03–2.70	1.19	0.62–2.31
	Culture 6	1.55	1.04–2.30	1.47	0.93–2.32	1.10	0.58–2.06

^1^ Phase 1, before multimodal intervention (January 2019 to 16 June 2021) and Phase 2, after multimodal intervention (January 2022 to 16 June 2024). ^2^ Phase 1, before multimodal intervention (January 2019 to 16 June 2021) and Moment 2, after blood culture pack (January 2023 to 16 June 2024). ^3^ Phase 2, Moment 1, before blood culture pack (January 2022 to December 2022) and Moment 2, after blood culture pack (January 2023 to 16 June 2024).

## Data Availability

For confidentiality purposes, the data are held by the corresponding author (S.F.) and will be made available upon reasonable request.

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
