# Peer review of "Effectiveness of A Nurse-Led Multimodal Intervention in Preventing Blood Culture Contamination: A Before-and-After Study"

_healthcare, 2024, doi:10.3390/healthcare12171735_

Round 1

Reviewer 1 Report

Comments and Suggestions for Authors

Overall, well written

The number of blood cultures should be 23,566 and not 23.566.

Kindly look for the punctuation marks errors, they are in results section too.

Study has been conducted in Wards. No ICU included ?

Comments on the Quality of English Language

-

Author Response

Please see the attachment (Section: Authors' responses to Reviewer 1).

Reviewer 2 Report

Comments and Suggestions for Authors

Dear authors,

The manuscript's topic is very current. It indicates the effects of a nurse-led multimodal intervention in a clinical setting to prevent blood culture contamination. The study uses a robust methodology that is rigorously explained and effectively supported visually.

I would like to make some suggestions that could improve the manuscript:

Introduction

The introduction clearly indicates the importance of taking an adequate blood sample for blood culture and the possibility and consequences of contamination in the pre-analytical phase. It also mentions the impact of this issue on the efficient, safe, and high-quality provision of health care. The statements in lines 46-48 are well-known and accepted by health professionals, for which there are several references. It is necessary for the statement to be supported by references. I suggest you correct/complete it.

Furthermore, the contribution of nurses and the factors that can influence the contamination rate of blood cultures are clearly indicated. It also indicates that previous studies have shown positive results from educational interventions and the introduction of standardized blood culture packages. It would be helpful to state where and for which areas/topics educational interventions were implemented and in what clinical environments standardized blood culture packages were applied. I suggest you provide these details.

Methods

Multimodal intervention, methods of collecting blood samples for blood culture, and criteria for contaminated blood culture are very precisely described. The visual presentation of the timeline and development of the multimodal intervention, which was simultaneously accompanied by supplementary material in the appendix and a three-minute video, made a special contribution to the methodology.

It would be helpful if you could provide more clarity on whether the pandemic period had any effect on the study, given that it was also conducted during that period. Also, in lines 149-150, you state the current evidence. Please support it with a reference.

Results

The results are presented in tables with adequate accompanying text.

I observed that you listed a total number of 7.42 cultures in Table 1 for Phase 2, Moment 2. Was this a technical data entry error? Please correct it.

Discussion

The discussion is extensive, and the authors draw attention to many important questions raised by their research and the results of other authors regarding the implementation of multimodal interventions, available resources, nursing practice and their impact on reducing the contamination rate of blood cultures.

Moreover, they clearly indicate limitations that may affect the limited generalization of the obtained data. At the same time, they indicate that future research should include a comprehensive collection and analysis of additional variables.

The conclusions are concise, well-argued, and based on research results. The references mentioned are relevant to the topic that the paper dealt with.

I hope you find my comments helpful.

Author Response

Please see the attachment (Section: Authors' responses to Reviewer 2).

Reviewer 3 Report

Comments and Suggestions for Authors

Thank you for inviting me to review this well-written paper. Although it was easy to understand, it did not actually provide any information or knowledge that was not already known by microbiologists and infection control nurses.

Line 47: the statement here can be expanded by mentioning that many contaminants (such as coagulase-negative Staph.) are also only susceptible to antibiotics with higher toxicity potential (such as vancomycin).

Line 57: BCC has to be spelt out because it is being encountered for the first time in the manuscript.

Line 75: Interesting…. how does end-stage renal disease or older age increase BCC rates?

Line 94: out of these 10,928 cultures, it has to be specified if they contained blood culture sets. If so, was each set counted as 1 or 2 cultures? The same comment applies to the 12,638 cultures on line 95. The logical approach is to consider each set as a single culture because any BCC will most likely affect both vials, artificially inflating the BCC figures.

Line 159: state the chemical composition of the solution used to perform hand hygiene.

The video uploaded to via www.powtoon.com was professionally done (although unfortunately I could not understand any of the words because they were not in English).

The blood culture pack detailed on line 195 appears to be intended for adult patients. For paediatric patients, was a separate pack containing the BacT/ALERT PF vial available? This question in important because on line 110, cultures from children were also included in the study.

Author Response

Please see the attachment (Section: Authors' responses to Reviewer 3).

Reviewer 4 Report

Comments and Suggestions for Authors

The paper “Effectiveness of a nurse-led multimodal intervention in preventing blood culture contamination: a before-and-after study” deals with two extremely important issues for nursing: patient safety and quality of care. It highlights contamination as a critical factor which, when properly prevented, leads to positive results, both by reducing antibiotic resistance and the complications associated with it.

The article is very well written and has the right methodological approach to the research question and objective.

The Title is clear and coherent with the research question, objective and content of the article.

The Abstract is well structured for the content of the manuscript, presenting the main elements of the study (objective, methodology, results and conclusions).

As far as the Keywords are concerned, I recommend that the authors rethink adding the terms “Patient safety” and “Quality improvement”, since the intervention carried out has a positive impact on the quality of care provided to patients and consequently improves patient safety and efficacy.

The Introduction frames the problem well and makes the link to the study's objective.

The Materials and Methods section is very well described, with perfectly clear methodological rigor. The study design used makes it possible to clearly assess the impact of the multimodal intervention on reducing blood culture contamination rates, and is therefore suitable for intervention studies in clinical settings.

The description of the multimodal intervention is very well described and detailed, making it very clear to the reader.

The Statistical Analysis section is correctly described, with all the necessary information.

The Results and Discussion sections are very well described, comparing results and discussing the results in the light of the evidence.

The Conclusions are also well described with reference to the issues of nurse training and the quality of nursing care.

The Bibliographical References are somewhat up-to-date, considering that of the 40 references used, 26 are more than 5 years old and 12 are more than 10 years old.

I would like to congratulate the authors on this important paper.

Author Response

Please see the attachment (Section: Authors' responses to Reviewer 4).
